# Combined Positive Score for Programmed Death Ligand-1 Expression and Inflammatory Microenvironment in Gastrointestinal Stromal Tumors

**DOI:** 10.3390/medicina58020174

**Published:** 2022-01-24

**Authors:** Vlad Herlea, Alexandra Roșulescu, Violeta Claudia Calotă, Vlad Croitoru, Elena Stoica Mustafa, Cătălin Vasilescu, Sorin Alexandrescu, Traian Dumitrașcu, Irinel Popescu, Simona Olimpia Dima, Maria Sajin

**Affiliations:** 1Department of Pathology, Fundeni Clinical Institute, 022328 Bucharest, Romania; alexandra.rosulescu@yahoo.com (A.R.); elenam_stoica@yahoo.com (E.S.M.); 2Faculty of Medicine, “Titu Maiorescu” University, 031593 Bucharest, Romania; irinel.popescu220@gmail.com; 3Center of Excellence for Translational Medicine, Fundeni Clinical Institute, 022328 Bucharest, Romania; 4“Carol Davila” University of Medicine and Pharmacy, 020021 Bucharest, Romania; catvasilescu@gmail.com (C.V.); stalexandrescu@yahoo.com (S.A.); traian.dumitrascu76@gmail.com (T.D.); maria_sajin@yahoo.com (M.S.); 5National Institute of Public Health, 050463 Bucharest, Romania; viocalota@yahoo.com; 6Department of Oncology, Fundeni Clinical Institute, 022328 Bucharest, Romania; vlad.m.croitoru@gmail.com; 7Center of General Surgery and Liver Transplantation, Fundeni Clinical Institute, 022328 Bucharest, Romania

**Keywords:** gastrointestinal stromal tumors, PD-L1, combined positive score, immune cells

## Abstract

*Background and Objectives*: GISTs are the most frequent type of mesenchymal neoplasm of the digestive tract. The prognosis is mainly determined by tumor dimensions, mitotic rate and location, but other less well-documented factors can influence evolution and survival. The immune microenvironment and checkpoint molecule expression were proven to impact the prognosis in different types of cancer. The aim of this study was to determine PD-L1 expression in GISTs and to evaluate the level of intratumoral immune infiltration in relation to prognostic variables and survival. *Materials and Methods*: Sixty-five GISTs diagnosed in the same institution between 2015 and 2018 were immunohistochemically tested for PD-L1 and evaluated using CPS. Immune cells were emphasized, with CD3, CD4, CD8, CD20 and CD68 antibodies and quantified. All data were processed using statistical tools. *Results*: The median age was 61 years (range, 28–78) and 36 patients (55.4%) were males. The location of the tumors was predominantly gastric (46%), followed by the small bowel (17%) and colorectal (6%). In addition, 11% were EGISTs and 20% were secondary tumors (11% metastases and 9% local recurrences). PD-L1 had a variable expression in tumor and inflammatory cells, with a CPS ranging from 0 to 100. Moreover, 64.6% of cases were PD-L1 positive with no significant differences among categories of variables, such as the age and the sex of the patient, tumor location, the primary or secondary character of the tumor, dimensions, mitotic rate, the risk of disease progression and tumor cell type. Immune cells had a variable distribution throughout the tumors. CD3+ lymphocytes were the most frequent type. CD20+ cells were identified in a larger number in tumors ≤5 cm (*p* = 0.038). PD-L1-positive tumors had a higher number of immune cells, particularly CD3+, CD20+ and CD68+, in comparison to PD-L1-negative ones (*p* = 0.032, *p* = 0.051, *p* = 0.008). Epithelioid and mixed cell-type tumors had a higher number of CD68+ cells. Survival was not influenced by PD-L1 expression; instead, it was decreased in multifocal tumors (*p* = 0.0001) and in cases with Ki67 ≥ 50% (*p* = 0.008). *Conclusions*: PD-L1-positive expression and the presence of different immune cell types, in variable quantities, can contribute to a better understanding of the complex interactions between tumor cells and the microenvironment, with a possible therapeutic role in GISTs.

## 1. Introduction

Gastrointestinal stromal tumors (GISTs) are the most common non-epithelial neoplasms of the digestive tract. They originate in the interstitial cells of Cajal, which are involved in the intestinal peristalsis and are located within the muscularis propria. The great majority of GISTs result from the activating of mutations in c-KIT or PDGFRA. Although classified as malignant, this tumor type has a wide spectrum of biological behavior, from clinically benign to highly aggressive, with recurrences and metastases [1,2]. Extragastrointestinal stromal tumors (EGISTs) are rare tumors, mostly located in the omentum, mesentery and retroperitoneum, with a similar histopathological aspect, immunohistochemical and biomolecular profile to GISTs. Their origin is unclear, with one hypothesis being that EGISTs are actually mural GISTs that lost their connection with the digestive tract wall. In comparison with gastric and intestinal GISTs, EGISTs are characterized by a poorer prognosis, have larger dimensions and are diagnosed at younger ages [3,4,5].

Risk stratification schemes that are nowadays used (modified NIH consensus criteria and AFIP-Miettinen criteria) take into account tumor location, tumor dimensions and mitotic rate [6,7]. Treatment consists in surgical resection and targeted therapy with tyrosine kinase inhibitors (imatinib, sunitinib) that are proven to prolong survival and delay the appearance of metastases; however, there are cases that lack treatment response. In this context, additional therapeutic strategies need to be developed for the cases that do not respond to currently available treatments [1].

Inflammatory cells associated with tumors have been, in recent years, investigated with the intent to discover their prognostic role. Additionally, the immune response could be therapeutically targeted, as in the case of several malignant tumors that currently benefit from specific immune treatments. Several studies have shown that a high level of intratumoral lymphocytes represents a favorable prognostic factor in different tumor types [8,9,10,11]. The microenvironment in GISTs includes immune cells in variable amounts. Studies investigating this subject are not numerous, but the results are encouraging. Most of the inflammatory cells in GISTs are represented by macrophages and T cells [11]. B lymphocytes and natural killer cells are rare in primary tumors, while in metastases, they seem to be present in a higher quantity [12].

Closely related to inflammatory cells is PD-L1, an immune checkpoint molecule that can be expressed in tumor-associated immune cells and also in tumor cells. The interaction between PD-1 and PD-L1 in the tumor microenvironment promotes immune tolerance. To prevent this, monoclonal antibodies (immune checkpoint inhibitors) were developed and, moreover, they have entered the clinical practice [13,14,15]. The clinical significance of PD-L1 expression in GISTs is not yet clearly understood, therefore further research is necessary to discover its role in tumor evolution.

The aim of this study was to investigate the PD-L1 expression in GISTs, according to the Combined Positive Score (CPS), and to evaluate the intratumoral immune cells by quantifying the number of lymphocytes and histiocytes. The results were analyzed in relation to the risk of disease progression and selected clinical parameters, including survival.

## 2. Materials and Methods

This study represents a retrospective analysis of 65 cases of gastrointestinal stromal tumors that underwent surgical resection in the Surgery Departments of Fundeni Clinical Institute (Bucharest, Romania) and which were diagnosed in the Pathology Department of the same institution between 2015 and 2018. The data regarding the pathological aspects were collected from the histopathological records of the cases, while details concerning clinical presentation and treatment were extracted from the hospital’s internal database. The patients involved in the research signed their informed consent, allowing the use of their tissues in scientific studies. Additionally, approval for the use of samples and data was obtained from local ethics committee.

The resected specimens were handled following the standard protocols that are being used in histopathology laboratories. After fixation in 10% buffered formalin, adequate fragments were selected in the course of macroscopic examination, and the tumor sections were processed and embedded in paraffin blocks. Afterwards, fine sections of 3–4 microns were realized and stained with hematoxylin and eosin (H&E). For each case, one representative tissue block was selected for subsequent tests.

The following aspects were collected and analyzed: age of the patient, gender, clinical signs and symptoms, tumor location, tumor dimensions, cellular type (spindled, epithelioid or mixed), mitotic rate (<5/50 HPFs or >5/50 HPFs), risk of disease progression (very low → high), prognostic group (1 → 6b), treatment with tyrosine kinase inhibitors, recurrences, metastases and survival.

Immunohistochemistry. All analyzed cases were immunohistochemically tested to confirm the diagnosis of GIST and to exclude the differential diagnoses (CD117, DOG1, CD34, SMA and S100). In the next stage, markers for PD-L1 and immune cells (CD3, CD4, CD8, CD20 and CD68) were performed, in an attempt to establish the relationship between tumor cells and the immune response. Additionally, a proliferation index (Ki67) was determined (see Table 1). To improve the accuracy of the evaluation, two pathologists evaluated and analyzed the immunohistochemical slides.

For the immune cell quantification, both in the H&E and IHC, 5 pictures with a high power field (40×) were taken per slide, with captures of representative aspects, and with the use of ImageJ program, the immune cells were counted on each field. Afterwards, the mean was calculated, resulting the number of total/specific immune cells that was assigned to every case.

The PD-L1 expression was analyzed and reported using the Combined Positive Score (CPS), which considers PD-L1 expression on tumor cells and also on immune infiltrating cells. The CPS is defined as the number of positive tumor cells, lymphocytes and macrophages divided by the total number of viable tumor cells, multiplied by 100 [16,17].

Statistical analysis. Data were charted and analyzed using Excel 2013 and STATA MP Version 13.0 (College Station, TX, USA), and the results we expressed as means ± standard deviation (SD), or median (range) for continuous variables, and as frequency and percentage for categorical variables. Significant differences in the number of immune cells were evaluated using the Wilcoxon–Mann–Whitney test. Associations between categorical variables were assessed using Pearson’s chi-square test or Fisher’s exact test. The relationship between overall survival and categorical variables was assessed with the Kaplan–Meier method and the log-rank test. The Cox regression model was used to perform univariate and multivariate survival analyses and to calculate hazard ratios. All statistical tests were two sided and a level of *p* < 0.05 was used to indicate statistical significance.

## 3. Results

### 3.1. Patient and Tumor Characteristics

A cohort of 65 patients diagnosed with GIST was studied. In total, 29 (44.6%) were females and 36 (55.4%) were males, with ages ranging between 28 and 78 years, with a mean of 56 for women and 59 for men. Tumors were located in the stomach in 46% of cases, followed by the small bowel in 17% of cases, and only 6% were colorectal. In addition, 11% were EGISTs and 20% were secondary tumors (11% metastases and 9% local recurrences). The tumors measured between 0.5 and 21 cm, with a mean of 7.1 cm for gastric tumors, 5 cm for small bowel tumors, 7.6 cm for colorectal tumors, 9 cm for EGISTs and 7.5 cm for secondary tumors. The mitotic rate was ≤5/50 HPFs in 60% of cases and >5/50HPFs in 40% of cases. The prognostic group among primary tumors was variable, ranging from 1 to 6b. Risk of disease progression was from very low to high: six patients (11.5%) were included in the very-low-risk group, 24 (46.2%) were in the low-risk group, six (11.5%) were in the moderate risk group and 16 (30.8%) were in the high-risk group. Morphologically, 63.1% of tumors had fusiform cellularity, 4.6% were composed of epithelioid cells and 32.3% were of a mixed type. Moreover, 92.3% were CD117-positive cases and value of Ki67 proliferation index varied between 1% and 75% (59 cases < 50%, six case types; see Table 2).

### 3.2. PD-L1 Expression

PD-L1 had a variable expression in tumor cells and in inflammatory cells (Figure 1). With a CPS cutoff of 1, 42 cases (64.6%) were PD-L1 positive, including 12 with a score of 1, 7 with score 2 and 23 with score 3, while 23 cases (35.4%) were PD-L1 negative. CPS values ranged from 0 to 100, with a median of 5.5 for gastric location, 12 for the small intestine, 50 for the colon, 40 for the rectum, 5 for EGIST and recurrences and 7 for metastases. No significant differences in PD-L1 expression were observed between distinct locations (*p* = 0.586), in cases with recurrences compared with the absence of recurrences (*p* = 0.331) and in cases with or without metastases (*p* = 0.776) (Table 3, Figure 2). The CPS had a median value of 5 in spindle cell GISTs, 7 in the epithelioid type and 21 in tumors with mixed cellularity (Figure 3).

No significant differences regarding PD-L1 expression were registered between categories in variables such as the sex and the age of the patient, tumor size, cell type, mitotic rate or the risk of disease progression (see Table 4).

### 3.3. Immune Cells

Immune cells were quantified with H&E on immunohistochemical slides (CD3+, CD20+, CD4+, CD8+ and CD68+ cells). Their intratumoral distribution was variable, from isolated to diffuse and in aggregates (Figure 4). CD3+ lymphocytes were the dominant type. Relative to tumor size, we found that tumors ≤5 cm had a larger number of CD20-positive cells (*p* = 0.038) when compared to larger tumors (Figure 5). CD68+ cells were in a larger amount in tumors > 5 cm, although this was not significant (*p* = 0.059). The other immune cell types did not quantitatively correlate with tumor dimensions.

The number of immune cells was higher in PD-L1-positive tumors in comparison with negative tumors (counted on H&E slides). Additionally, a higher number of CD3+, CD20+ and CD68+ cells correlated with PD-L1-positive tumors (*p* = 0.032, 0.051, respective *p* = 0.008) (see Table 5). The number of CD68+ cells was higher in epithelioid and mixed cell types (*p* = 0.004). No other statistically significant correlations were established.

### 3.4. Survival Analysis

Overall survival was significantly shorter for patients with multifocal tumors compared with those with monofocal tumors (*p* = 0.0001). Tumors with a high Ki67 value (≥50%) are associated with a decreased overall survival (*p* = 0.008; see Table 6). After comparing the survival in patients with PD-L1-positive and PD-L1-negative tumors, no statistically significant differences were observed between the two groups (*p* = 0.492, log-rank test; see Figure 6). Adjusting PD-L1 according to risk of disease progression and cell dimensions maintained this result (*p* = 0.579; see Table 7). A multivariate analysis that included tumors and Ki67 showed that only the presence of multifocal tumors significantly reduced the survival (OR = 4.66, 95% CI:1.81–11.96, *p* = 0.001). Other variables, such as gender and tumor location, were not proven to impact the survival (Table 6).

## 4. Discussion

The tumor microenvironment is composed of various type of cells, such as fibroblasts, macrophages, lymphocytes and myeloid-derived suppressor cells, and also of an extracellular matrix with structural and functional roles. It enables tumor progression by inducing immune suppression, inhibiting the apoptosis of neoplastic cells and stimulating angiogenesis [18].

In recent years, the immune microenvironment in tumors has generated great concern, as it represents the target of novel therapeutic agents for cases refractory to conventional treatment. Additionally, immune checkpoint inhibitors, proved to be effective in treating several malignant tumors, thus increasing the interest of exploring the expression of molecules such as PD-L1 in more and more cancer types. Several studies have proved that the immunohistochemical evaluation of PD-L1 in tumor cells is a good predictor of response to treatment with PD1-PD-L1 inhibitors [19,20]. In cancers such as melanoma, lung and genitourinary carcinomas, patients with PD-L1-positive tumors had a significantly higher chance of responding to treatment. In melanoma, PD-L1 expression was associated with a good prognosis, while in renal cell carcinoma and non-small cell carcinoma, it was associated with a worse prognostic [21]. In GIST, some authors proved that PD-L1 is an independent prognosis factor, with a low expression being associated with a higher risk of metastasis [22,23].

In our study, PD-L1 expression was variable, being focal in the majority of cases, similar to the results of other researchers [24]. When quantifying PD-L1 expression, we used the CPS. It considers the immunolabeling both of the tumor cells (TPS) and of the immune cells (MIDS). This method was developed after studies had shown that TPS alone is not a useful predictive biomarker for treatment response and MIDS is not easily reproducible, although it is useful. These two methods combined proved to be a robust and reproducible modality for scoring PD-L1 in tumors [16,25].

Applying CPS, in our case series, we found that 64.6% were PD-L1-positive tumors. A study carried out on PDGFRA mutant GISTs showed that PD-L1 expression was higher in epithelioid cell types and in tumors with small dimensions, suggesting that it is associated with a better prognosis [26]. Our study showed no correlation between PD-L1 and tumor cell type, tumor dimensions and risk of disease progression (Table 4). We also compared PD-L1 expression in primary tumors, different locations and secondary tumors (recurrences and metastases), the latter representing more advanced stages of disease, but no significant differences were found.

In relation to checkpoint molecules are the inflammatory cells, which were also analyzed in our research. Studies have shown that in a hypoxic tumor microenvironment, the immune checkpoints, including PD1, PD-L1 and CTLA 4, are upregulated, as are the immunosuppressive cells. In such a state, effector T cells are inhibited and suppressive cells, such as regulatory T cells and tumor-associated macrophages, are in an increased amount [27]. In our samples, intratumoral immune cells were present in variable amounts. The dominant immune cell type was represented by T lymphocytes, similar to the results of Pantaleo et al. [28]. Additionally, Cameron et al. found that T lymphocytes are the dominant type in primary GIST and that the number of T cells is superior in metastases [29]. Likewise, other authors reached the conclusion that the quantity of immune cells is different in metastatic tumors compared to those in non-metastatic tumors, and that this is influenced by the stem-like properties of the tumor cells [30]. These findings emphasize the importance of inflammatory cells in GIST microenvironments and their role in tumor progression.

A higher number of T and B lymphocytes correlated with small tumor dimensions (Figure 3). We also found that PD-L1-positive tumors have higher amounts of immune cells, particularly CD3+, CD20+ and CD68+ cells. Related to this topic, Zhao et al. discovered that a PD1/PD-L1 blockade rescues exhausted T cells, enhancing the response to classical treatment [22].

GIST prognosis is mainly influenced by tumor dimensions, mitotic rate and location [3,4]. Given that tumor evolution is determined by complex interactions between cancer cells, check point molecules and immune cells, multiple biomarkers should be investigated for determining if they influence the prognosis and survival. In our research, immune infiltration did not correlate with survival. By comparison, in intestinal-type gastric cancer, elevated numbers of CD3+ and CD8+ cells were associated with increased overall survival [9]. Additionally, PD-L1 expression did not impact the survival. Instead, multifocality and a proliferation index Ki67 > 50% was associated with a reduced overall survival.

These findings suggest that certain GIST categories could qualify for immune mediated therapies, but further investigation is necessary.

We consider that our study is important in the search for therapeutic solutions for GISTs that resist the available lines of treatment. Immunotherapy is a promising solution, and studies that explore the immune microenvironment in this tumor entity are few in number and ask for further investigation [31,32,33]. The assets of the present paper are that it evaluates both the PD-L1 expression and immune infiltrate in GIST, it brings to attention a more accurate method of scoring PD-L1 (i.e., CPS) and it analyses different tumor risk classes. The differences between our results and those of the other researchers mentioned may be, in part, due to the methods that were used, distinct demographics, the dissimilar size of the studied cohorts and small samples bringing an increased risk of biased conclusions. Nevertheless, we have the belief that our results will contribute to a better understanding of the complex interactions that take place between tumor cells and the microenvironment in this tumor entity.

## 5. Conclusions

PD-L1-positive GISTs are infiltrated by a higher number of immune cells, particularly CD3+, CD20+ and CD68+ cells. Tumors with smaller dimensions have more CD20+ cells. CD68+ cells are present in a larger number in tumors > 5 cm, and the number of CD68+ cells was higher in epithelioid and mixed cell types. PD-L1 and inflammation does not influence overall survival, which is affected instead by tumor multifocality and a high Ki67 proliferation index.

## Figures and Tables

**Figure 1 medicina-58-00174-f001:**
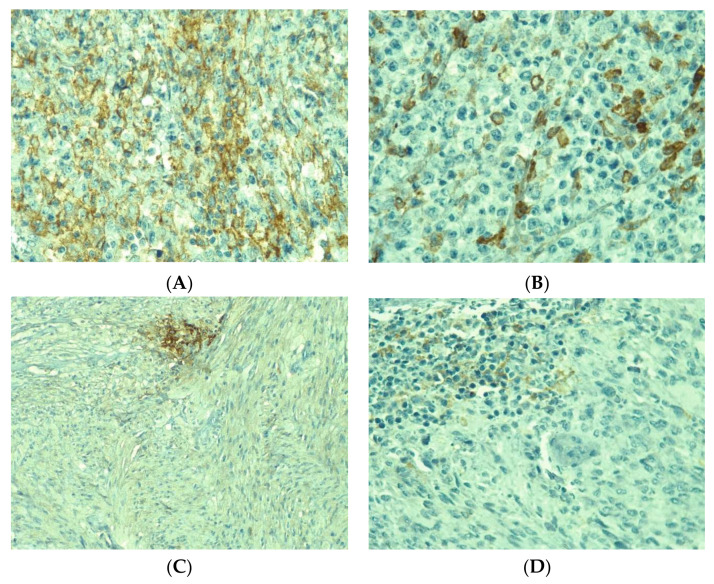
PD-L1 expression in GIST. (**A**) Diffuse positive tumor cells with membrane staining (CPS > 50)–40×. (**B**) Sparse positive tumor cells (CPS >10)–40×. (**C**) Focal positive tumor cells (CPS > 1)–20×. (**D**) Focal positive immune cells (CPS > 10)–40×.

**Figure 2 medicina-58-00174-f002:**
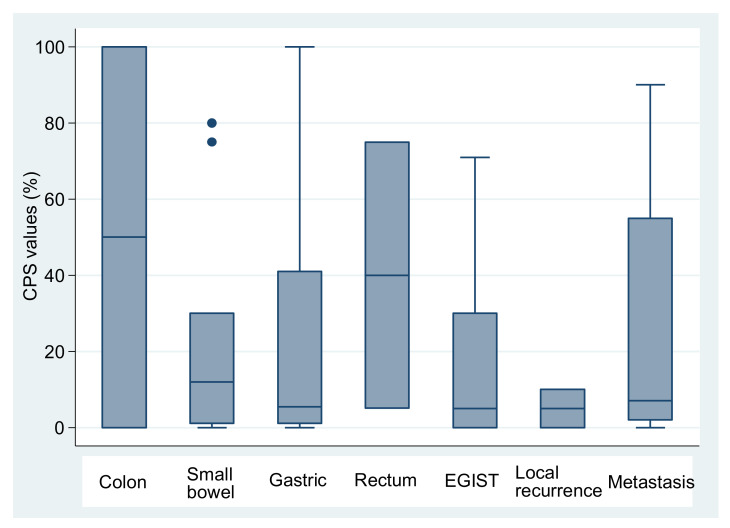
CPS values by disease site. Mean ± SD of CPS for Colon (N = 2): 50 ± 70.7; Small bowel (N = 11): 22.3 ± 29.4; Gastric (N = 30): 23.9 ± 31.6; Rectum (N = 2): 40 ± 49.5; EGIST (N = 7): 16.8 ± 25.9; Local recurrence (N = 6): 5 ± 5.5; Metastasis (N = 7): 25.6 ± 34.3. The outliers are presented as dots.

**Figure 3 medicina-58-00174-f003:**
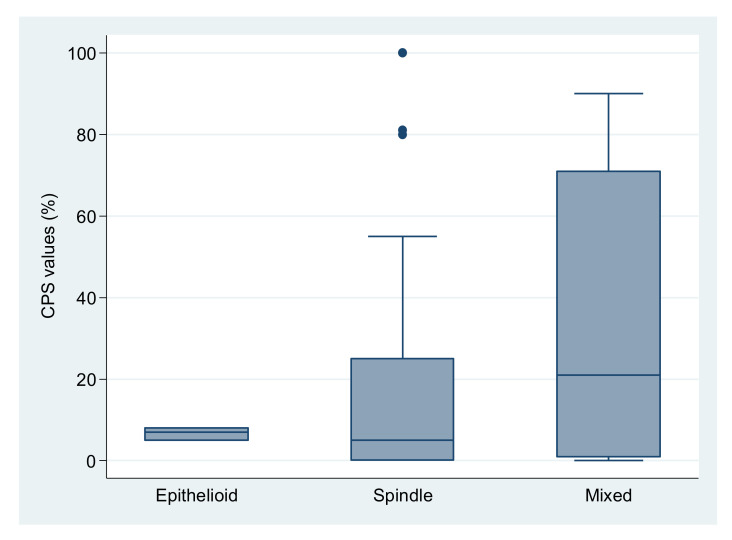
CPS values for different tumor cell types. Mean ±SD: Epithelioid (N = 3): 6.7 ± 1.5; Spindle (N = 41): 18.4 ± 28.4; Mixed (N = 21): 33.1 ± 35.0). The outliers are presented as dots.

**Figure 4 medicina-58-00174-f004:**
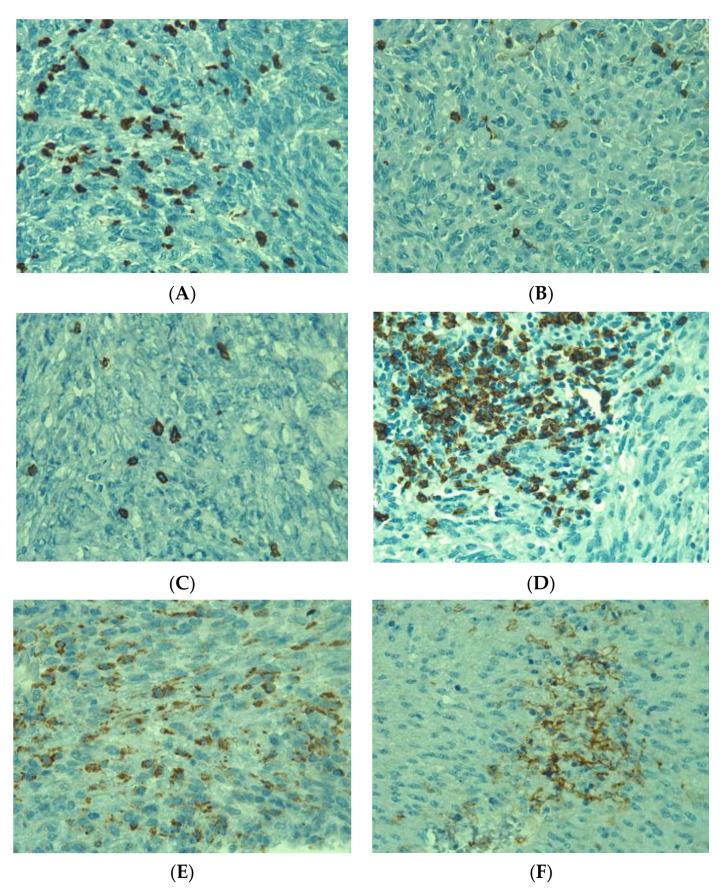
Immunohistochemical expression of intratumoral immune cells in gastrointestinal stromal tumors (40×): (A) CD3+ cells diffusely infiltrating among tumor cells, (B) scattered CD4+ cells, (**C**) sparse infiltration of CD8+ cells, (**D**) aggregate of CD20+ cells, (**E**) Diffusely distributed CD68 cells, (**F**) aggregate of PD-L1-positive cells.

**Figure 5 medicina-58-00174-f005:**
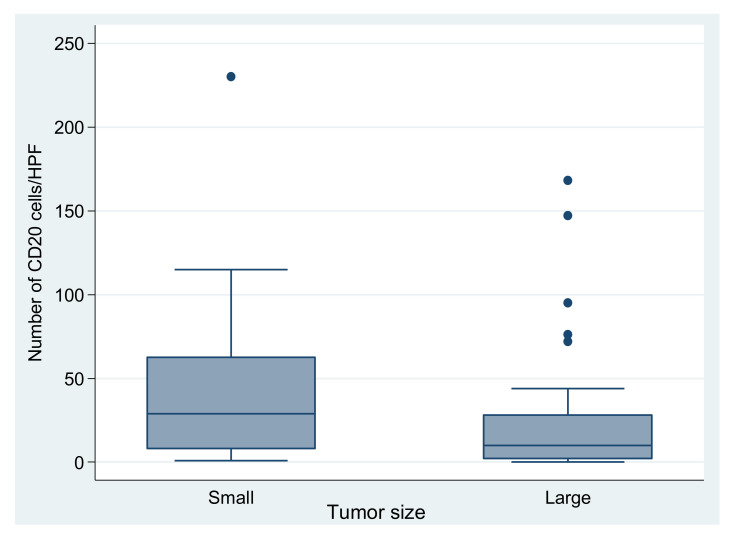
CD20+ cell quantification relative to size of the tumor (small= ≤5 cm, N = 32 (49,2%), large >5 cm, N = 33 (50.8%); mean ± SD of CD20: 40.9 ± 46.9 in smaller vs 26.6 ± 41.3 in larger tumors; *p* = 0.038). The outliers are presented as dots.

**Figure 6 medicina-58-00174-f006:**
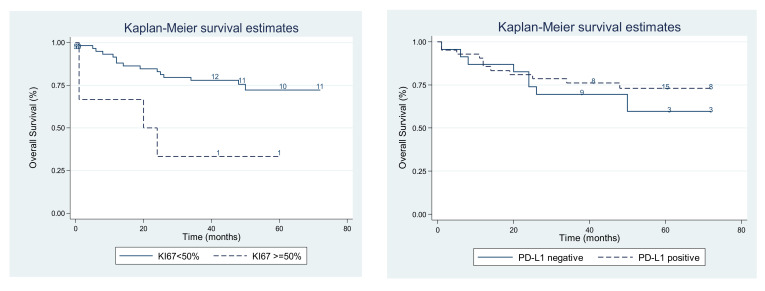
Survival relative to: **left**: Ki67 proliferation index; **right**: PD-L1 expression.

**Table 1 medicina-58-00174-t001:** The details of the immunohistochemical markers used in tumor microenvironment analysis.

Antibody	Dilution	Clone	Producer
PD-L1	1:50	ZR3	Cell Marque (USA)
Ki67	1:250	SP6	Cell Marque (USA)
CD3	Ready to use	MRQ-39	Cell Marque (USA)
CD20	1:250	L26	Cell Marque (USA)
CD4	1:50	SP35	Cell Marque (USA)
CD8	1:50	C8/144B	Cell Marque (USA)
CD68	1:250	Kp-1	Cell Marque (USA)

**Table 2 medicina-58-00174-t002:** Summary of the clinicopathologic features of the studied cohort.

Patient Characteristic	*N* (%)
**Age**, mean (sd)	57.8 (11.4)
**Gender**	
Female	29 (44.6)
Male	36 (55.4)
**Tumor location**	
Colon	2 (3.1)
Small bowel	11 (16.9)
Gastric	30 (46.1)
Rectum	2 (3.1)
EGIST	7 (10.8)
Local recurrence	6 (9.2)
Metastasis	7 (10.8)
**Tumor dimension** (cm), mean (sd)	7.1 (4.8)
Colon	10 (2.8)
Small bowel	5.0 (2.8)
Gastric	7.2 (5.0)
Rectum	5.2 (6.7)
EGIST	9 (6.3)
Local recurrence	8.7 (3.0)
Metastasis	6.5 (6.1)
**Risk of disease progression**	
Very low	5 (9.8)
Low	24 (47.1)
Moderate	6 (11.8)
High	16 (31.4)
**Mitotic rate**	
≤5/50 HPFs	39 (60.9)
>5/50 HPFs	25 (39.1)
**Cell type**	
Spindle	41 (63.1)
Epithelioid	3 (4.6)
Mixed	21 (32.3)
**CD117**	
Negative	5 (7.7)
Positive	60 (92.3)
**Ki67** (range)	1–75
<50%	59 (90.8)
≥50%	6 (9.2)

**Table 3 medicina-58-00174-t003:** PD-L1 expression (CPS value) related to location, in primary and secondary tumors.

	PD-L1 CPS				*p*
Location of the Tumor N (%)	≤1 24 (36.9)	2–10 17 (26.1)	11–50 12 (18.5)	>50 12 (18.5)	
Colon	2 (100)	0	0	0	0.586
Small bowel	5 (45.3)	2 (18.9)	2 (18.9)	2 (18.9)	
Gastric	9 (30.0)	9 (30.0)	4 (13.3)	8 (26.7)	
Rectum	1 (50.0)	0	0	1 (50.0)	
EGIST	3 (42.9)	3 (42.9)	1 (14.2)	0	
Local recurrence	1 (16.7)	1 (16.7)	3 (50.0)	1 (16.7)	
Metastasis	3 (42.8)	2 (28.6)	2 (28.6)	0	
Recurrence					
No	19 (33.9)	14 (25.0)	11 (19.6)	12 (21.5)	0.331
Yes	5 (55.6)	3 (33.3)	1 (11.1)	0	
Metastasis					
No	19 (39.7)	11 (22.9)	9 (18.7)	9 (18.7)	0.776
Yes	5 (29.5)	6 (35.3)	3 (17.6)	3 (17.6)	

**Table 4 medicina-58-00174-t004:** PD-L1 expression in different clinical and pathological categories of GIST.

Factors	Negative PDL-1	Positive PDL-1	*p* *
**Sex, N(%)**			
Females	11 (37.9)	18 (62.1)	0.796
Males	12 (33.3)	24 (66.7)	
**Age**			
Mean ± sd	54.3 ± 10.7	59.7 ± 11.4	0.068 **
Median (range)	57 (35–72)	62 (28–78)	
**Tumor size, N (%)**			
≤5 cm	10 (31.2)	22 (68.8)	0.606
>5 cm	13 (39.4)	20 (60.6)	
**Cell type, N (%)**			
Epithelioid and mixed	8 (33.3)	16 (66.7)	1.000
Spindle	15 (36.6)	26 (63.4)	
**Mitotic rate, N (%)**			
≤5	15 (38.5)	24 (61.5)	0.431
>5	7 (28.0)	18 (72.0)	
**Risk of disease progression, N (%)**			
Very low and Low	12 (41.4)	17 (58.6)	0.566
Intermediate and High	7 (37.2)	15 (62.8)	

* Fisher’s exact test; ** Student’s *t*-test.

**Table 5 medicina-58-00174-t005:** Immune cells number relative to PD-L1 expression.

Cell Type (*n* = 65)	PDL-1 Negative (*n* = 23)	PDL-1 Positive (*n* = 42)	*p* *	All Patients
	Median(range)-No.of cells/HPF			
CD3	21 (3–95)	37 (5–217)	0.032	31 (3–217)
CD4	28 (3–109)	41.5 (0–119)	0.091	36 (0–119)
CD8	13 (0–119)	23 (4–161)	0.066	19 (0–161)
CD20	7 (0–230)	15 (0–168)	0.051	13 (0–230)
CD68	21 (2–80)	38 (2–222)	0.008	34 (2–222)
H&E	124 (31–460)	172.5 (42–729)	0.011	148 (31–729)

* Wilcoxon–Mann–Whitney test.

**Table 6 medicina-58-00174-t006:** Association of clinicopathological features with overall survival.

	*N* (%)	No. Events	Median OS	*p* *
**Gender**				
Female	29 (44.6)	9	48	0.711
Male	36 (55.4)	10	36	
**PDL-1**				
Negative	23 (35.4)	8	36	0.492
Positive	42 (64.6)	11	48	
**KI67**				
<50	59 (90.8)	15	48	0.008
≥50	6 (9.2)	4	22	
**Multifocal tumor**				
No	48 (73.8)	8	48	0.0001
Yes	17 (26.1)	11	25	
**Location of the tumor**				
Colon	2 (3.1)	0	42	0.301
Small bowel	11 (16.9)	4	60	
Stomach	30 (46.1)	8	36	
Rectum	2 (3.1)	1	18.5	
EGIST	7 (10.8)	4	25	
Recurrences	6 (9.2)	1	66	
Metastases	7 (10.8)	1	60	

* Log-rank test.

**Table 7 medicina-58-00174-t007:** The impact of PDL-1 adjusted for tumor size and risk of disease progression on overall survival.

Factors	HR (95% CI)	*p*-Value *
**Risk of disease progression** Very low and Low Intermediate and High	Reference 4.46 (0.64–30.90)	0.130
**Tumor size** ≤5 cm >5 cm	Reference 2.30 (0.33–16.08)	0.400
**PDL-1** Negative Positive	Reference 0.71 (0.21–2.38)	0.579

* Multivariate Cox model, N = 51 observations.

## Data Availability

Not applicable.

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
