# Peer review of "Combined Positive Score for Programmed Death Ligand-1 Expression and Inflammatory Microenvironment in Gastrointestinal Stromal Tumors"

_medicina, 2022, doi:10.3390/medicina58020174_

Round 1

Reviewer 1 Report

Herein, Drs. Herlea and colleagues perform an analysis of 65 patients with GIST for PD-L1 expression and characterization of immune cell infiltrate. While the integrity of this work cannot be doubted, I unfortunately have significant concerns related to the biologic significance of this study, as well as the way the data are presented. Significant changes are needed to convince the reader of the justification for the study, contextualize the importance of the study’s results, as well as to more clearly and succinctly present the data to the reader.

Major:

  • The abstract is lacking in useful data. Suggest rewriting to focus on specifying methods and focusing on detailing the key findings with numeric values.
  • The introduction is unfocused and lacks thematic progression or a clear biological justification why the study was performed given that in >90% of GISTs highly effective first-line TKI therapies exist, with 2nd 3rd and even 4th line therapies currently available. Additionally, PD-L1 has been previously evaluated in GIST by
  • Please include a table detailing the clinicopathologic features of the studied cohort, with N(%) for each variable rather than just percent as is frequently listed in 3.1
  • Figure 1 and Table 2’s design is confusing. A histogram with median, IQR, and outliers by disease site is more appropriate and could replace both for more succinct data assimilation.
  • Please justify analyzing PD1 expression on recurrences and metastases alongside primary tumors. The biology is different here and grouping them together is not ideal.
  • Figure 2 is confusing and would similarly be better represented as a histogram. The maximum Y axis should be 100 to reflect the maximum CPS score rather than 120.
  • Table 4 and figure 3 are largely redundant, recommend selecting one. Also they lack units of measurement. Are these cells per HPF?
  • All figures require additional information in their legend so that the reader can understand them standalone from the rest of the article.
  • Figure 7 and table 6 are insufficient to evaluate PD-L1 association with survival. A multivariable analysis including tumor size, Grade, location, etc. is needed, as these are important covariates. Statistical consultation is recommended
  • The discussion is unfocused and tangential and it is difficult to glean the authors’ points.

Minor:

  • Table 5 lacks units of measurement
  • Please include number at risk for each group as a separate table embedded at the bottom of figure 7.

Reviewer 2 Report

This an interesting study concerning Programmed Death Ligand expression and inflammatory microenvironment in Gastrointestinal Stromal Tumors.

The manuscript is well developed

I have the following comments:

  • The period of the study must be added in the abstract
  • Moderate English-revision required for the whole manuscript
  • Please add both the limitations and the clinical implication of the study
  • The discussion of the Tumor Microenvironment needs to be expanded: 1) Hypoxia-Related Gene-Based Signature Can Evaluate the Tumor Immune Microenvironment and Predict the Prognosis of Colon Adenocarcinoma Patients. Int J Gen Med. 2021 Dec 16;14:9853-9862. doi: 10.2147/IJGM.S3432162) Immune Infiltration, Cancer Stemness, and Targeted Therapy in Gastrointestinal Stromal Tumor. Front Immunol. 2021 Dec 3;12:691713. doi: 10.3389/fimmu.2021.6917133)Therapeutic Targets and Tumor Microenvironment in Colorectal Cancer. J Clin Med. 2021 May 25;10(11):2295. doi: 10.3390/jcm10112295

Round 2

Reviewer 1 Report

I applaud the authors for their attempt to improve upon their manuscript. Unfortunately significant issues remain that prevent me from recommending acceptance at this time:

Major:

  • Figure 1 is not the appropriate figure type to represent the data and is now redundant with figure 2. Figure 2 is closer to what is needed, but the Y axis is unlabeled. Please justify the inclusion of both figures or remove Figure 1.
  • I am still unclear as to what figure 4 is trying to show. The Y axis is unlabeled. The word “distribution” implies a geographic component, such as at different locations within the tumor. Perhaps the term “presence of inflammatory cells” would be clearer? Also there is some overlap with Table 5 in terms of what is being reported, as the two columns combined would constitute the values for figure 4, if I am understanding correctly. I would again suggest trying to simplify where possible, such as by adding an “all patients” column to Table 5 and eliminating figure 4, unless the authors feel the graphical representation is crucial.
  • PDL-1 now appears in Table 6 twice with two separate P values, with one now controlling for tumor size, risk of progression, and location of tumor. This is confusing, because I had assumed that the log-rank test was being used for this table given that number of events and median OS was reported. However, controlling for these other variables would require a cox proportional hazards model. Please specify the statistical test used for this table in a footnote, they should all be the same for a table looking at survival. If choosing log-rank, it is not possible to control for these other variables. If choosing Cox modeling, data should be given as hazard ratios with 95% confidence intervals. It is also important to give the HRs and Cis for the adjusting variables if you built a multivariable model for this.
  • Most importantly, given that this is largely a negative study with respect the association of PD-L1 expression in GISTs with the measured outcomes (survival, immune cell populations), the argument for why this study is important is largely missing. Simply because something has not been looked at before does not make a study significant or worthy of publication. In my prior review I mentioned that PD-L1 has been previously evaluated in GIST. One such reference is “PDL1 expression is an independent prognostic factor in localized GIST” by Bertucci and colleagues in 2015. This manuscript actually came to different conclusions from Bertucci and colleagues with respect to survival. The rationale for the present study and the findings of the present study should be discussed in light of this and all other papers on this subject (a literature search must be done and appears to have been omitted here). It is crucial to contextualize the results of research with what has come before. The authors must answer the following questions:
    • Why was this study important to do given the results of prior studies? Were they conflicting or incomplete somehow in their analysis?
    • Why do the present study’s results differ from prior work? Provide an objective assessment of the risk of bias/erroneous conclusions in both your and prior studies on this topic given that this conflict exists.

Minor:

  • Formatting issues remain with the introduction. One paragraph is just one sentence, and another does not have an indent.
  • Formatting issues exist with Table 3 (border lines)
  • Table 4: Please use Fisher’s exact test rather than Chi Squared test for comparisons involving a 2x2 table (sex, tumor size, mitotic rate, and cell type (while epithelioid is combined with mixed into one group)
  • The discussion contains grammatical and punctuation errors.
  • As a general rule, paragraphs should not be one sentence. This occurs several times in the manuscript

Author Response

Dear Madam or Sir,

Thank you for giving us the opportunity to submit a revised draft of the manuscript “Combined positive score for Programmed Death Ligand – 1 expression and inflammatory microenvironment in Gastrointestinal Stromal Tumors” for publication in the medical journal Medicina. We appreciate the time and effort that you and the reviewers dedicated to providing feedback on our manuscript and are grateful for the comments and suggestions of improvement to our paper. The suggestions and comments have been closely followed and revisions have been made accordingly. The following are the questions extracted from the reviewers’ comments along with our summarized responses.

  1. Figure 1 is not the appropriate figure type to represent the data and is now redundant with figure 2. Figure 2 is closer to what is needed, but the Y axis is unlabeled. Please justify the inclusion of both figures or remove Figure 1.

Author response: We removed figure 1

  1. I am still unclear as to what figure 4 is trying to show. The Y axis is unlabeled. The word “distribution” implies a geographic component, such as at different locations within the tumor. Perhaps the term “presence of inflammatory cells” would be clearer? Also there is some overlap with Table 5 in terms of what is being reported, as the two columns combined would constitute the values for figure 4, if I am understanding correctly. I would again suggest trying to simplify where possible, such as by adding an “all patients” column to Table 5 and eliminating figure 4, unless the authors feel the graphical representation is crucial.

Author response: We removed figure 4 and added the column in table 5

  1. PDL-1 now appears in Table 6 twice with two separate P values, with one now controlling for tumor size, risk of progression, and location of tumor. This is confusing, because I had assumed that the log-rank test was being used for this table given that number of events and median OS was reported. However, controlling for these other variables would require a cox proportional hazards model. Please specify the statistical test used for this table in a footnote, they should all be the same for a table looking at survival. If choosing log-rank, it is not possible to control for these other variables. If choosing Cox modeling, data should be given as hazard ratios with 95% confidence intervals. It is also important to give the HRs and Cis for the adjusting variables if you built a multivariable model for this.

Author response: We made the changes according to your counsel

  1. Most importantly, given that this is largely a negative study with respect the association of PD-L1 expression in GISTs with the measured outcomes (survival, immune cell populations), the argument for why this study is important is largely missing. Simply because something has not been looked at before does not make a study significant or worthy of publication. In my prior review I mentioned that PD-L1 has been previously evaluated in GIST. One such reference is “PDL1 expression is an independent prognostic factor in localized GIST” by Bertucci and colleagues in 2015. This manuscript actually came to different conclusions from Bertucci and colleagues with respect to survival. The rationale for the present study and the findings of the present study should be discussed in light of this and all other papers on this subject (a literature search must be done and appears to have been omitted here). It is crucial to contextualize the results of research with what has come before. The authors must answer the following questions:
    • Why was this study important to do given the results of prior studies? Were they conflicting or incomplete somehow in their analysis?
    • Why do the present study’s results differ from prior work? Provide an objective assessment of the risk of bias/erroneous conclusions in both your and prior studies on this topic given that this conflict exists.

Author response:  We made a supplementary literature search and presented our point of view towards the importance of our results, in comparison with the existing data

  1. Formatting issues remain with the introduction. One paragraph is just one sentence, and another does not have an indent.

Author response: We made the corrections

  1. Formatting issues exist with Table 3 (border lines)

Author response:  We made the corrections

  1. Table 4: Please use Fisher’s exact test rather than Chi Squared test for comparisons involving a 2x2 table (sex, tumor size, mitotic rate, and cell type (while epithelioid is combined with mixed into one group)

Author response: We made the changes according to your counsel

  1. The discussion contains grammatical and punctuation errors.

Author response: We made additional corrections

  1. As a general rule, paragraphs should not be one sentence. This occurs several times in the manuscript

Author response: You are right. We made the corrections.

Reviewer 2 Report

I'm satisfied with the changes made

Author Response

Thank you very much for your comments and advices